



Using data and model to infer climate and environmental changes during the
Little Ice Age in tropical West Africa
Anne-Marie Lézine[1], Maé Catrain[1], Julián Villamayor[1,2] and Myriam Khodri[1].
1. Laboratoire d'Océanographie et du Climat. Expérimentation et Approche numérique/IPSL.
Sorbonne Université-CNRS-IRD-MNHN. 4 Place Jussieu. 75005. Paris. France
2. Department of Atmospheric Chemistry and Climate, Institute of Physical Chemistry
Rocasolano, CSIC, Madrid, Spain.
**Abstract**
Here we present hydrological and vegetation paleo-data extracted from 28 sites in West Africa
from 5° S to 19° N and the past1000/PMIP4 IPSL-CM6A-LR climate model simulations covering
the 850-1850 CE period to document the environmental and climatic changes that occurred
during the Little Ice Age (LIA). The comparison between paleo-data and model simulations
shows a clear contrast between the area spanning the Sahel and the Savannah in the North,
characterized by widespread drought, and the equatorial sites in the South, where humid
conditions prevailed. Particular attention was paid to the Sahel, whose climatic evolution was
characterized by a progressive drying trend between 1250 and 1850CE. Three major features
are highlighted: (1) the detection of two early warning signals around 1170 and 1240CE
preceding the onset of the LIA drying trend; (2) an irreversible tipping point at 1800-
1850CE characterized by a dramatic rainfall drop and a widespread environmental
degradation in the Sahel; and (3) a succession of drying events punctuating the LIA, the major
of which was dated around 1600CE. The climatic long-term evolution of the Sahel is associated
with a gradual southward displacement of the Inter-Tropical Convergence Zone induced by
the radiative cooling impacts of major volcanic eruptions that have punctuated the last
millennium.

**1.  Introduction**
Precipitation in tropical West Africa is closely related to the West African Monsoon (WAM)
system, created by the temperature land-sea contrast between the tropical Atlantic and the
west of the African continent (Nicholson, 2013). The WAM long-term variability during the
20[th] century has focused much attention due to the severe consequences in the Sahel semi-
arid region, which experienced a long period of drought in the 1970-80s (Folland et al. 1986;
Giannini et al. 2003). It is broadly accepted that these changes were mainly driven by the sea
surface temperature (SST) variability (Folland et al. 1986; Mohino et al. 2011; Rodríguez-
Fonseca et al. 2015), amplified by land surface processes (Giannini et al. 2003; Kucharski et al.
2013). However, only a few works document the WAM variability prior to the 20th century
(Nicholson et al. 2012; Gallego et al. 2015; Villamayor et al. 2018) due to the little information
covering the 19[th] century and beyond. The paleo-archives are rare, often incomplete, and
suffer from often poorly constrained chronologies. Moreover, these archives are rarely direct
records of climate parameters, but indirect ones, namely historical, biological, or
sedimentological.  They integrate not only changes in environmental parameters but also the
vital effect of species, the vulnerability or the resilience of ecosystems and the cultural
adaptations of populations. Here we use pollen and other environmental proxies as well as

[1] Corresponding author : anne-marie.lezine@locean.ipsl.fr



historical chronicles to document the last millennium with a special focus on the period from
1250 to 1850 CE including the transition between the Medieval Climate Anomaly (MCA; 950-
1250CE) and the Little Ice Age (LIA; 1450-1850CE) periods characterised by global
temperatures respectively above and below average (Nash et al. 2016; Villamayor et al. sub.).
The aim of this review is not to record the climate variability at interannual scale but to discuss
the timing, distribution and magnitude of the major secular environmental changes which
punctuated the LIA in northern tropical Africa with a focus on the regional biomes and
hydrological systems responses times to rainfall anomalies.
**2. Material and method**
**2.1 Paleo-data**

This paper uses compilations of paleo-records from different sources with the highest
available resolution (Table 1; Fig. 1). These data have the advantage of providing continuous
records over the last millennium, but their temporal resolution is generally mostly
(multi)decadal to centennial : pollen data are used for vegetation reconstructions (Elenga
1992 ; Reynaud-Farrera et al. 1996; Ballouche 1998; Vincens et al. 1998; Salzmann et al. 2005;
Ngomanda et al. 2007; Waller et al. 2007; Brncic et al. 2009; 2017; Lézine et al. 2011; 2013;
2019; Lebamba et al. 2016; Tovar et al. 2019; Fofana et al. 2020; Catrain 2021), and
micropaleontological, sedimentological and geochemical data to capture hydrological and
climatic changes (Bertaux et al. 1998 ; Holmes et al. 1999 ; Street-Perrott et al. 2000 ; Schefuss
et al. 2005 ; Wang et al., 2008 ; Shanahan et al. 2009 ; Mulitza et al. 2010 ; Nguetsop et al.
2010 ; 2011 ; 2013 ; Carré et al. 2019 ; Lézine et al. 2019 ; Fofana et al. 2020 ; Catrain 2021).
Compilations of historical chronicles (Nicholson 1978; 1980; 2013; Nicholson et al. 2012;
Coquery-Vidrovitch 1997; Maley and Vernet 2013) and intrumental records (Gallego et al.
2015) have also been examined, although the first are based on records of extreme events
only (droughts, floods) and the second are limited in their temporal coverage. All these data
are also scattered in a few limited areas of the Sahel (Senegal, Southern Mauritania, Niger
River inner loop, Lake Chad basin) with possible redundancies.
The resulting data set is highly heterogeneous. Therefore, the data have been homogenized
as follows: (1) only records covering the interval between 900 CE and present day with at least
a 100-year temporal resolution have been taken into account, (2) in order to evaluate the
relative amplitude of the environmental/climate change, we build a 6-point scale ranging from
0, corresponding to the most degraded environment (e.g., drying of lakes, salinization of
water, increase of dust transport, degradation/opening of the vegetation cover) or the driest
climate, up to 6, which refers to the less degraded environment (e.g., high lake level, fresh
water, dense vegetation cover) or the wettest climate.

| Site name | proxy | latitude | longitude | reference | Sector/vegetation zones |
|---|---|---|---|---|---|
| Lake Yoa | Pollen/sediment | 19.057621 | 20.500690 | Lézine et al. 2011 | Sahara (Desert) |
| GeoB9501 | Dust fraction | 16.83333 | -16.73333 | Mulitza et al. 2010 | Sahel |
| St Louis | Pollen/Diatom | 16.03508 | -16.48382 | Fofana et al. 2020 | Sahel (grasslands and wooded grasslands) |





| Mboro-Baobab | Pollen/Diatom | 15.149132 | -16.909275 | Lézine et al. 2019 | Sahel (grasslands and wooded grasslands) |
|---|---|---|---|---|---|
| Oursi | Pollen | 14.65283 | -0.486 | Ballouche 1998 | Sahel (grasslands and wooded grasslands) |
| Dioron Boumak | Geochemistry | 13.835809 | -16.498372 | Carré et al, 2019 | Sahel/Savannah boundary |
| Lake Jikaryia | Sediment/Mineral-magnetic | 13.3136667 | 11.077 | Waller et al. 2007; Wang et al. 2008 | Sahel (grasslands and wooded grasslands) |
| Lake Bal | Ostracods/Chemistry | 13.304 | 10.943 | Holmes et al. 1999 | Sahel (grasslands and wooded grasslands) |
| Lake Kajemarum | Dust fraction/Geochemistry | 13.303 | 11.024 | Street-Perrott et al. 2000 | Sahel (grasslands and wooded grasslands) |
| Lake Chad | Historical | 13.053472 | 14.463469 | Maley and Vernet 2013 | Sahel (grasslands and wooded grasslands) |
| Lake Mbalang | Pollen/Diatoms | 7.316 | 13.733 | Vincens et al. 2000; Nguetsop et al. 2011 | Savannah |
| Lake Tizong | Pollen/Diatoms | 7.25 | 13.583 | Nguetsop et al. 2013; Lebamba et al. 2016 | Savannah |
| Lake Sélé | Pollen | 7.15 | 2.433 | Salzmann et al. 2005 | Savannah |
| Lake Bosumtwi | Geochemistry | 6.5 | -1.416 | Shanahan et al. 2009 | Central Africa (lowlands) (Equatorial forests) |
| Mbi | Pollen | 6.089273 | 10.348549 | Lézine et al., in press | Central Africa (highlands) (Afromontane forests) |
| Lake Bambili | Pollen/ Geochemistry | 5.936 | 10.242 | Lézine et al. 2013 | Central Africa (highlands) (Afromontane forests) |
| Lake Petpenoun | Pollen | 5.64147 | 10.64531 | Catrain 2021 | Savannah |





| Lake Ossa | Pollen/Diatoms | 3.800 | 10.75 | Reynaud Farrera et al. 1996; Nguetsop et al. 2010 | Central Africa (lowlands) (Equatorial forests) |
|---|---|---|---|---|---|
| Mopo Bai | Pollen/Geochemistry | 2.240 | 16.261388 | Brncic et al. 2009 | Central Africa (lowlands) (Equatorial forests) |
| Bemba Yanga | Pollen | 2.18726 | 16.52513 | Tovar et al. 2019 | Central Africa (lowlands) (Equatorial forests) |
| Goualougo | Pollen | 2.0875 | 16.54722 | Brncic et al. 2017 | Central Africa (lowlands) (Equatorial forests) |
| Lake Nguène | Pollen | −0.2 | 10.466 | Ngomanda et al. 2007 | Central Africa (lowlands) (Equatorial forests) |
| Lake Kamalété | Pollen | -0.7166 | 11.7666 | Ngomanda et al. 2007 | Central Africa (lowlands) (Equatorial forests) |
| Lake Sinnda | Pollen/Sediment | -3.836111 | 12.8 | Bertaux et al. 1996 ; Vincens et al. 1998 | Central Africa (lowlands) (Equatorial forests) |
| Ngamakala | Pollen | -4.075 | 15.38333 | Elenga 1992 | Central Africa (lowlands) (Equatorial forests) |
| Lake Kitina | Pollen/Sediment | -4.27 | 12 | Bertaux et al. 1996 ; Elenga et al. 1996 | Central Africa (lowlands) (Equatorial forests) |
| GeoB6518-1 | Alkenone / Geochemistry | -5.588333 | 11.221667 | Schefuss et al. 2005 | Central Africa |


Table 1: Geographical positions, type and references of paleo-records used in this study (see
Fig. 1).

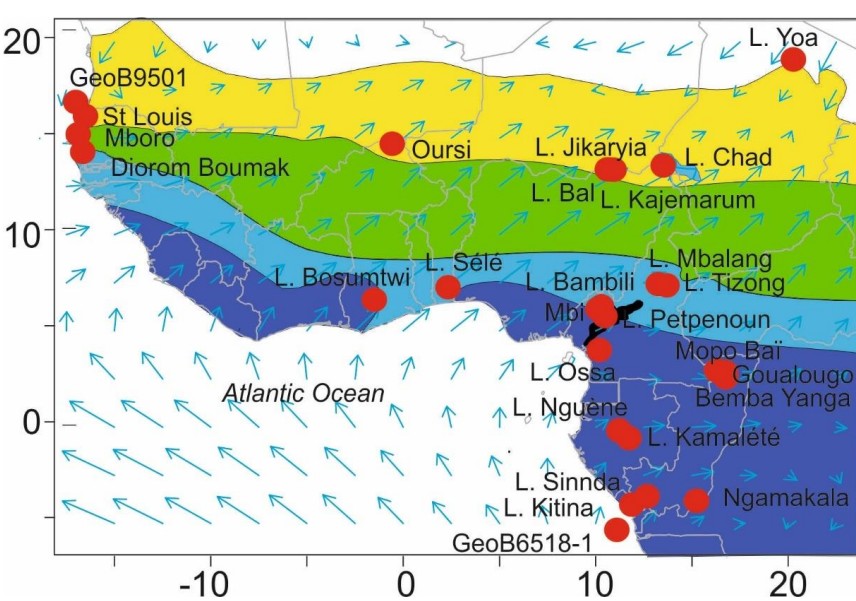

**Figure 1:** Map showing the location of paleorecords available in tropical West Africa documenting the last millennium (Table 1). Blue arrows indicate the strength and direction of the main 925 hPa monsoonal winds during boreal summer, i.e., the WAM rainy season (NCEP-DOE AMIP-II Reanalysis (Kanamitsu et al., 2002)). In color, vegetation units from White (1983): dark blue: Guineo-Congolian rainforest; light blue: Sudano-Guinean woodland and wooded grassland (here referred to as Savannah (vegetation) zone); green: Sudanian woodland and wooded grassland ; yellow: Sahelian grassland and wooded grassland. black: Afromontane forest.

In order to verify whether the methodology employed is realistic and provides reliable indications of environmental change for the period prior to the instrumental records, scores of the WAM rainy season (July to September) multidecadal hydrological changes from natural archives and historical data (Table 1) in the Sahel are compared to the African Southwesterly Index (ASWI) developed by Gallego et al. (2015) over 1840-1990 CE. The ASWI was validated against instrumental observations as a good measure of WAM intensity during the rainy season over the instrumental period (Gallego et al. 2015). Positive values of the ASWI indicate periods when the monsoon is well established over the Sahel, and thus define periods of heavy rainfall in the region, which is consistent with observational data (Descroix et al., 2015). Figure 2 shows that our historical records give a magnitude of dry and wet anomalies that reflects the sensitivity of populations to periods of drought or flooding. Our assessment of hydrological conditions based on natural archives reflects historical records variations but with a somewhat weaker magnitude. This is probably due to the much lower temporal resolution of the available data (25-50 yrs on average). It is also worth noting that the lake data corresponds to a precipitation/evaporation balance and not the precipitation amounts at a given site. Nevertheless, the curves are remarkably similar and point to wet periods centred ca 1875 and 1950 CE.





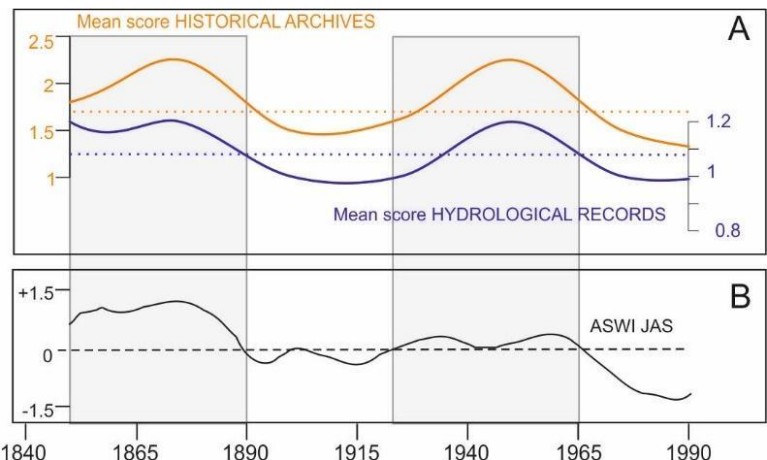

115
116

**Figure 2:** Observed and reconstructed rainfall anomalies over the Sahel during the 1840-1990 CE period. (A) the mean scores from historical (yellow curve) and natural archives (blue curve) for the Sahel (Nicholson, 1978; 1980; Nicholson et al. 2012; 2013; Coquery-Vidrovitch, 1997; Holmes et al., 1999; Street-Perrott et al. 2000; Waller et al. 2007; Wang et al. 2008; Mulitza et al. 2010; Maley and Vernet, 2013; Lézine et al. 2019). The dotted yellow and blue lines correspond respectively to the historical and paleohydrological archives mean scores during the period 1850-1990CE. They allow identifying anomalously wet and dry periods. (B) The African Southwesterly Index (ASWI) developed by Gallego et al. (2015) as a measure of rainfall anomalies in Sahel during the WAM rainy season (July to September).

**2.2 Model experiments**

In this study we compare reconstructed environmental changes in Western Africa to those simulated in the past1000 model experiment covering the 850-1850 CE climate performed as part of 4[th] phase of the Paleoclimate Modelling Intercomparison Project (PMIP4; Jungclaus et al. 2017; Kageyama et al. 2017) by the IPSL-CM6A-LR model version developed for the Coupled Model Intercomparison Project phase 6 (CMIP6) at Institut Pierre-Simon Laplace (Boucher et al. 2020; Lurton et al. 2020). The IPSL-CM6A-LR model couples the atmospheric component LMDZ (Hourdin et al. 2020) to the land surface model ORCHIDEE (d'Orgeval et al., 2008) and to the ocean model NEMO, which includes other models to represent sea-ice interactions (Rousset et al., 2015) and biogeochemistry processes (Aumont et al. 2015). The atmospheric and land-surface grid have a resolution of 2.5° in longitude and 1.3° in latitude with 79 vertical layers. The oceanic component has 75 vertical levels with a mean spatial horizontal resolution of about 1° and a refinement of 1/3° near the equator. This model reproduces fairly well the ENSO seasonality despite the sea surface temperature anomalies extending too westward in the central Pacific during El Niño events. The spatial pattern of the AMV teleconnection in the Pacific is consistent with observations but the tropical Atlantic variability is relatively weaker. Unlike most current state-of-the-art CMIP6 models, the IPSL-CM6A-LR model simulates a predominant secular variability in the Atlantic with AMV peaks separated by about 200 years (Boucher et al., 2020).

The past1000 IPSL-CM6A-LR model experiment is designed to simulate the climate response to natural forcings recommended by PMIP4 (Jungclaus et al. 2017) and covering the pre-

industrial millennium (850-1849CE), namely the time varying astronomical parameters, the
trace gases (Meinshausen et al. 2017; Matthes et al. 2017), the eVolv2k volcanic forcing
(Toohey and Sigl 2017), the SATIRE-M 14C solar activity with an adaptation of the spectral
irradiance to the CMIP6 *historical* forcing and the land use forcing (Lawrence et al. 2016).
Three past1000 IPSL-CM6A-LR model simulations have been performed and branched off from
various initial conditions in a 600 years long spinup run with fixed external radiative forcing to
the year 850 CE. This spinup run, itself branched off from the IPSL-CM6A-LR pre-industrial
control (piControl) run with constant external radiative forcing, has been performed to avoid
any spurious drift in the past 1000 experiments that could be related to the adjustment of the
slow components of the climate system (such as the ocean), to the different radiative balance
at the beginning of the last millennium as compared to the pre-industrial levels.

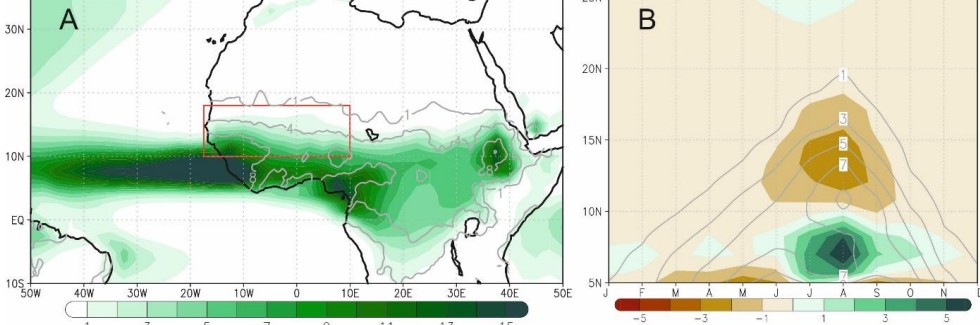

**Figure 3:** Climatological bias of simulated monthly precipitation. A) JAS mean averaged across
(colors) the 2000 year piControl run and (contours) the 1891-2019 period in GPCCv2020
observational database. B) (colors) Meridional seasonal cycle of the 10º W – 10º E mean model
bias (simulation minus observations) compared to (contours) the GPCPv2020 climatology. All
units are mm/day. Red box in (A) indicates the Sahel region (17.5ºW-10ºE; 10º-18ºN).
The IPSL-CM6A-LR model reproduces the observed climatological distribution of maximum
rainfall across West Africa during the WAM rainy season (Fig. 3A). The timing of the simulated
WAM seasonal cycle is also in good agreement with observations, with a well-defined onset
of the rainy season in July and then a demise after September (Fig. 3B). However, the
northward shift of maximum rainfall over the Sahel (north of 10ºN) during the rainy season is
slightly underestimated by the model, resulting in a climatological rain belt over West Africa
that is slightly more constrained to tropical regions compared to observations and dryer Sahel
on average. However, the well-characterized WAM seasonal timing suggests that there are no
remarkable biases affecting the simulated precipitation variability.
Then, to characterize the simulated Sahel rainfall variability over the past millennium and
contrast to the reconstructed environmental series, an index is performed using the past1000
ensemble-mean precipitation anomalies (relative to the piControl average) from July to
September (JAS), area-weighted and averaged across the Sahel region (red box in Fig. 3A). In
order to highlight the variability at decadal to longer time scales, the index is low-pass filtered
using a 10-year moving mean. In turn, the ensemble-mean anomalies help highlight the
changes in precipitation induced by external forcing, common across the different



realizations, against internal variability. Therefore, the ensemble-mean past1000 index of
Sahel precipitation represents the variability at decadal to longer time scales modulated by
internal processes and external natural forcings, such as the radiative forcing induced by large
volcanic eruptions.
**3. Results**
**3.1 The hydrological records**
The hydrological records provide a contrasting picture from one region to another: the Sahel,
the Sudano-Guinean Savannah zone and the tropical forests. They also reveal some local
exceptions. As already noted (e.g., Vincens et al. 1999), the local hydrogeological context may
strongly affect the individual response of lakes and wetlands to rainfall variations and partly
explains this apparent heterogeneity.
The main characteristics of the hydrological evolution in the Sahel, in the Savannah zone and
in low and high altitude equatorial forests can be summarized as follows (Fig. 4):
• Data from the central and western Sahel (Fig. 4A) point to a relatively dry period at the
end of the first millennium (900CE) at Bal, Kajemarum and in the Senegal River
watershed (GeoB9501). A wet period followed, already present at Mboro near the
littoral, which lasted up to 1350CE. Except at Kajemarum and Jikarya, where
hydrological conditions remained relatively stable, a gradual trend toward increased
aridity is recorded in two steps dated ca. 1625CE and 1800CE, respectively. Then,
during the last two centuries, only minor fluctuations occurred in a general context of
widespread aridity.
In the lake Chad area, Maley and Vernet (2013) depict a rather different and complex
history probably due to the variety of the archives they used (both historical and
natural) and also to the complexity of the hydrology of this immense water body
(Pham-Duc et al. 2020) fed by underground waters and by rivers of distant
geographical origin. The authors identify two major periods of flooding in the lake
Chad area: from the onset of the millennium to ca. 1200CE, then between 1600 and
1700CE, with a series of dry periods in between then from 1700CE onwards.
• Only three sites document the hydrological evolution of the Savannah zone south of
the Sahel (Fig. 4C). These sites are located in the centre of the savannah zone (White
1985): two crater lakes on the Adamawa plateaus (Mbalang and Tizong) and the other
at the mouth of the tributary of Lake Petpenoun in the Grassfields region of Cameroon.
The Adamawa lakes do not show any significant hydrological changes throughout the
last millennium. In contrast Petpenoun records a clear evolution towards aridity which
started ca. 1425CE and culminated ca. 1650CE up to the present day.
• Diorom Boumak (Fig. 4B) is situated at the southern boundary of the Sahel, in the
littoral mangrove of the Saloum estuary. In contrast to the other sites from the Sahel
and savannah zone this site records a remarkable wet period between 1500CE and
1800CE. As elsewhere however, aridification started ca. 1800CE.
• The equatorial region is characterized by contrasting hydrological situations (Fig. 4E).
Low lake levels are recorded at Bosumtwi, Mopo Bai, Goualougo, Nguène-Kamálété
during a period centred around 1100CE in contrast to Sinnda and Kitina where moist
conditions occurred. Moisture increased as soon as 1350CE at Goualougo and
continued up to 1400CE at Mopo Bai and Kitina. Then, there is a clear opposition
between Sinnda, Nguène-Kamálété, Bosumtwi and Ossa where low lake levels



occurred during a dry phase between ca 1350 and 1700CE and Mopo Bai, Goualougo
and Kitina which are characterized by wetter conditions. In any case, the marine record
at the mouth of the Congo River (GeoB6518-1) suggests that all these hydrological
variations in the equatorial lowlands remained of relatively low amplitude.
In the Cameroon highlands (Fig. 4D), hydrological conditions steadily declined as
shown at lake Bambili, starting from ca. 1250 and culminating ca. 1675CE. This gradual
trend is interrupted ca. 1500CE by a more pronounced phase of lake level lowering.
The Mbi swamp displays a rather different pattern: here, the water level was relatively
low throughout the whole last millennium except to two discrete wetter phases ca.
1450 and 1650CE.



**Figure 4:** Mean scores of hydrological and vegetation changes along a North-South transect from the northern limit of the Sahel (Yoa) to the Congo basin (GeoB6518-1). Data are grouped within the phytogeographical entities defined by White (1983) in tropical Africa : Sahelian grassland and wooded grassland, Sudano-Guinean savannah, highland Afromontane forest, lowland Guineo-Congolian forest.



### 3.2 Pollen data

- In the open landscapes of the Sahara, Sahel and Savannah zones, vegetation changes were of minor amplitude except at sites where gallery forests were previously well developed. It is in the westernmost part of the Sahel that the most profound changes in vegetation cover are recorded : In the Niaye area (Mboro) and in the Senegal river delta (St Louis), the degradation of the landscape originated ca. 1300CE and accelerated ca. 1600CE to a maximum reached ca. 1850CE (Fig. 4F). A discrete vegetation recovery is then recorded in the 19th century. In contrast, sites from the central Sahel (Oursi and Jikaryia) remained relatively stable throughout the last millennium in spite of a slight degradation recorded at Oursi ca. 1050CE. North of the Sahel (Yoa), the aridification of the desert landscape accelerated from the 19th century onward. South of the Sahel, in the Savannah zone, lakes Tizong and Sélé do not record any marked environmental change contrary to Petpenoun where a slight degradation is recorded ca. 1425CE (Fig. 4G). At Mbalang, a discrete phase of vegetation recovery occurred between ca 1400-1600CE.
- The forest cover remained roughly unchanged in the central forest massif (Mopo Baï, Bamba Yanga. Goulalougo, Fig. 4I). In the western regions however, (Ngamakala, Kitina, Lake Ossa, Nguène and Kamalété) a trend toward forest development started ca. 1250-1350CE. In the Cameroon highlands (Fig. 4H), the forest development occurred later, ca 1550-1500CE, after a phase of forest clearance from 1000 to 1450CE.

### 3.3 Model results

The index of the ensemble-mean Sahel JAS precipitation simulated over the past millennium reveals a change from a relatively wet mean state in the MCA (950-1249 CE) to a drier one in the LIA (1450-1849) (Fig. 5), suggesting a shift of the average WAM rainfall regime. Such continuous decline presents a linear rate of the seasonal Sahel rainfall of -0.7 mm per decade over 1250-1849CE, resulting in a 7% loss of the mean precipitation in the LIA relative to MCA (Fig. 5). Regarding decadal variations, the ensemble-mean index of past1000 Sahel precipitation almost doubles its variability in the LIA with respect to the MCA (the variance in 859-1249CE is 51% higher than in 1450-1849CE), which suggests a more unstable rainfall regime, apart from drier on average, by the late past millennium in response to natural external forcings. As shown by Villamayor et al. (2022), such a simulated long term drying trend and increased LIA Sahel precipitation decadal variability is related to the volcanic forcing influence on SSTs, which integrates the induced radiative cooling (Fang et al. 2021). The more frequent large volcanic eruptions during the LIA, as compared to the MCA, is integrated by the ocean long memory, leading to a gradual SST decrease that is more pronounced in the Northern Hemisphere than the Southern Hemisphere. The relative North Atlantic SST cooling trend along 850-1849CE, gradually promotes a southward shift of the Inter-Tropical Convergence Zone (ITCZ) and a weakening of monsoon moisture inflow to Western Africa. The long term WAM weakening is further amplified in the few years following any new large volcanic event, which occurrences are indicated by the vertical dotted lines on Figure 5A. As a consequence, more frequent negative rainfall anomalies lasting at least 5 consecutive years are also evident during the LIA as compared to the MCA, with significant drying that can persist up to 60 years around clusters of eruptions such as those of the 19th century.




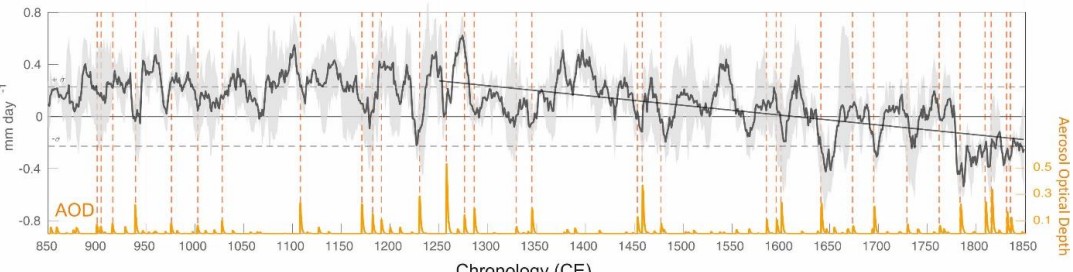


**Figure 5:** Multidecadal Sahel rainfall variability in IPSL-CM6A-LR past1000 simulations. Black
line: 10-years low pass filtered index of anomalous JAS Sahel precipitation anomalies averaged
in boxed area in Figure 3 (i.e., 10º-18ºN and17.5ºW-10ºE). The black line corresponds to the
ensemble mean, the grey shading to the ensemble spread and diagonal line to the 1250-1849
CE linear fit. Dashed horizontal lines show the +/-standard deviation of the equivalent
piControl index. The volcanic forcing used in the IPSL-CM6A-LR model experiments is shown
by the orange curve as the globally averaged Aerosol Optical Depth (AOD). Red vertical dotted
lines indicate the occurrence of strong volcanic eruptions about the size or larger that the
Pinatubo eruption (June 1991) defined when the tropical (20°S-20°N) or northern extratropical
(50°N-90°N) mean AOD is larger than 0.1.

**4. Discussion**

**4.1 Hydrology and Climate changes at secular timescale**

Data and past1000 model simulations show a strong North-South contrast between the Sahel
and Savannah zones, both subjected to severe drying during the LIA, and the equatorial areas,
spanning the Gulf of Guinea coast, suggesting an overall change of the WAM.



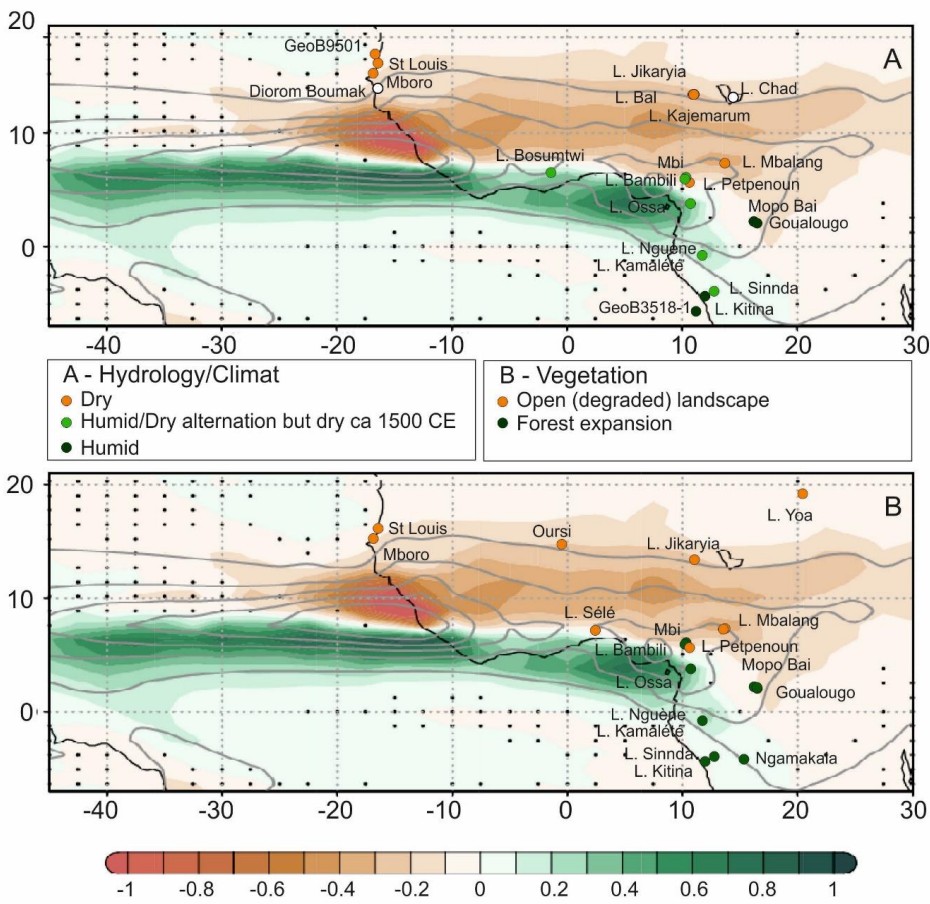

**Figure 6:** Distribution of JAS rainfall anomalies between the LIA (1450-1849 CE) and the MCA (950-1249 CE) as simulated by the IPSL-CM6A-LR model (shading, mm day$^{-1}$) compared to hydrological/dust (A) and vegetation (B) paleorecords during the LIA shown as dots following the same color scale as simulated anomalies. Grey contours indicate the piControl climatology from 2 mm day$^{-1}$ in intervals of 4 mm day$^{-1}$. Stippling indicates areas where the anomalies do not significantly emerge from the internal noise with 95% confidence level.

The difference between the simulated past1000 JAS precipitation during the LIA and the MCA shows a characteristic distribution of a weakened WAM associated with a southward shift of the ITCZ, with less rainfall across the Sahel and more in the Gulf of Guinea coast (Fig. 6). These simulated anomalies are consistent with the overall distribution of hydrological and vegetation proxy reconstructions.

**4.1.1 Hydrology**

Three major regions can be recognized from the paleohydrological records: The Sahel and Savannah zones, with drying trend; the center of the Congo Basin, which exhibit an opposite trend of increasing humidity; and the boundary between the dry and humid domains defined



by the equatorial sectors closest to the coast or in mountain, where an alternation of wet and
dry phases is recorded.  Two paleo-records differ from this general picture: that of Lake Chad,
where a period of flooding is recorded ca 1600CE, and that of the Diorom Boumak, where the
LIA is entirely characterized by a wet period (Fig. 4). As evoked above, the multiple origins of
the data and the complex hydrological system of Lake Chad may have introduced a bias into
the hydrological record and may explain (at least partly) the difference with the other Sahelian
archives. It is also likely that the rivers that feed the lake, which originate from southern
regions (the Chari and Logone rivers and their tributaries), may have caused an influx of water
during the short humid phase recorded on the Cameroon highlands (Bambili and Mbi) ca
1600CE. The case of the Diorom Boumak site is more complex: the historical records
mentioned by Maley and Vernet (2013) or Carré et al. (2019), among others, indicate that the
Saloum sector was wetter than the rest of the Sahel during part of the 16th century, allowing
for the establishment of two harvests per year. This may have been due, according to Maley
and Vernet (2013), to the occurrence of two rainy seasons, one in summer linked to the WAM
and the other in winter due to intense « Heug » rains linked to northern depressions. Carré et
al. (2019) however do not consider any other cause than the intensification of the
WAM.  Although the origin of this humid LIA in the Saloum still remains unexplained, the date
of its end, ca. 1800 CE, is consistent with all the other paleohydrological records from the
Sahel.
### 4.1.2 Vegetation
In the central Sahel, already degraded prior to the LIA (Lézine 2021), such as at Oursi, no
significant change occurred in the vegetation landscape which remained open throughout the
last millennium (Fig. 4B). The same pattern is observed in the wettest areas of the Congo Basin,
where the forests remained unchanged in composition and physiognomy (Tovar et al. 2019).
Elsewhere in the forest galleries of the Sahel and the Savannah zone (Mboro, St. Louis,
Petpenoun) the evolution of vegetation mirrored that of hydrological conditions while
recording a gradual degradation that culminated around 1800-1850 CE. In the westernmost
sector of the Sahel (Mboro, St Louis), the data suggest however a slight recovery of the
vegetation cover during the last few decades.
In contrast, both high and low elevation sites from the equatorial forest regions show an
opposite trend with marked forest recovery that began in the early years of the LIA and
accelerated around 1450CE. The forest expanded in the Equatorial lowlands despite increased
human presence has already been noted by Vincens et al. (1999). That means that the local
hydrological variations, and particularly the 1500 CE dry event, were of too small an amplitude
to impact forest dynamics. At most, a plateau in forest recovery is observed at that time
(Nguène, Kamalété). While the forest recovery was gradual at low altitudes, it seems to have
occurred more abruptly in the highlands.
### 4.2 The chronology of events at multidecadal timescale: focus on the Sahel and Savannah zone

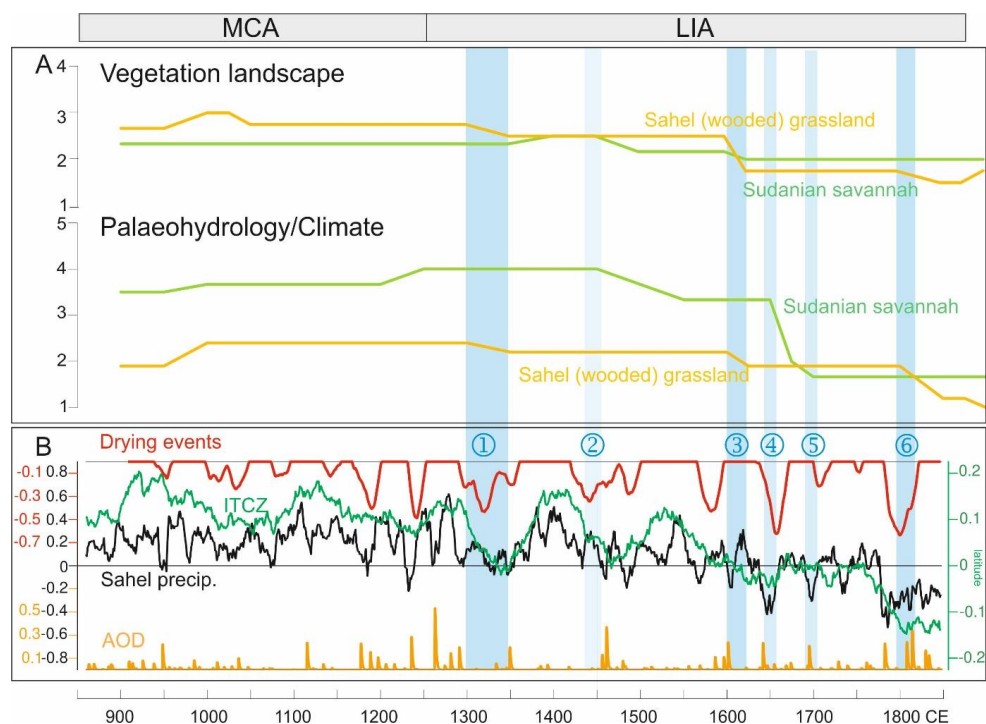

**Figure 7:** Multiproxy records of hydrology and vegetation during the last millennium in the
driest biomes (Sahel and Savannah zone) in western Africa (A) and long-term evolution of
rainfall over the Sahel as simulated by the IPSL-CM6A-LR past1000 model (B). Panel B: (Black
line) 10-year filtered ensemble-mean Sahel precipitation index (mm day$^{-1}$). (Green line) 50-
year filtered anomalous latitudinal position of the JAS ITCZ (defined as the latitudinal
maximum zonal-mean rainfall in 40ºW-10ºE) in the past1000 simulations respectively to the
piControl JAS mean position (in degrees of latitude). (Orange line) Global-mean AOD (volcanic
forcing). (Red line) Sahel Drying Persistence Index defined as the 50-year running negative
trend values over the Sahel ensemble-mean JAS precipitation index (mm day$^{-1}$ per 50 years).
Blue bars and numbers highlight the main climate/environmental degradation thresholds
identified in the paleo-records.

Environmental changes in the Sahel and Savannah zones during the LIA occurred in the context
of widespread environmental degradation that followed the severe environmental crisis at
the end of the African Humid Period (AHP; deMenocal et al., 2000). Between 3300 and 2500
cal yr BP (Lézine, 2021), the forests and woodlands, that widely expanded across the plains
and mountains of West Africa, strongly declined. This is particularly striking along the Atlantic
coast of Senegal, between 15° and 17° N where specific environmental conditions related to
the proximity of the sea and the presence of a water table near the surface favored the
development of exceptionally dense forest galleries of humid tropical affinity during the AHP
(Lézine 1989). As a result of this major environmental crisis, the Sahel and Savannah zone took
on its modern aspect of semi-desert grassland and wooded grassland. In this context,
discernible environmental fluctuations, particularly in vegetation, are of limited magnitude,
with the exception of sectors where forest galleries were widely established during the AHP.





To discuss the chronology of events that punctuated the LIA, paleo-data were averaged in
each geographical area (Sahel, Savannah zone) in the two categories covered by our study :
hydrology/climate and vegetation (Fig. 7A). A Drying Persistence Index was constructed from
our model results in order to quantify the Sahel precipitation deficit over at least 50-year
periods (red curve in Fig. 7B). It is defined at each year as the negative linear trend of the Sahel
ensemble-mean JAS precipitation (black curve in Fig. 7B) across the 50 previous years. We use
50 years to be consistent with the multi-decadal to centennial temporal resolution of the
paleo-data.
The past1000 simulations represent several drying events of various amplitude and duration
during the MCA that do not correspond to any major change in the vegetation of the Sahel
and Savannah zone. Instead, the environment in these two areas appears to be characterized
by a relatively stable humid regime (Fig. 7A).  This is coherent with the rainy mean state
represented by the past1000 simulations over the MCA, which is associated with an
anomalous northward ITCZ position (green curve in Fig. 7B) all over this period compared to
the LIA.
At the end of the MCA, two early warning signals (Lenton 2011) of Sahel drying events centred
at 1170 and 1240 CE are identified in our model experiments. The intensity and brevity of
these two events contrast with the minor dry phases identified prior to the LIA since the onset
of the last millennium. The Drying Persistence Index at these two events, which timing
coincides with the occurrence of large clusters of volcanic eruptions (orange curve Fig. 7B),
reaches over -0.3 mm day$^{-1}$ across 50 years. Both events preceded the onset of the LIA gradual
drying trend starting at 1250CE. This drying trend was sustained by the southward migration
of the ITCZ which shifts south of the piControl mean position at 1600 CE. It is consistent with
the continuous degradation of hydrological and vegetation conditions since 1250 CE in the
Sahel and Savannah zone identified in our multi-proxy records.
Several abrupt drying events larger than those identified during the MCA punctuated the LIA,
some of which reaching over -0.5 mm day$^{-1}$ across 50 years. Despite the difference in temporal
scale between the two approaches used here, there is a striking agreement between the major
simulated droughts and the environmental degradation steps in our paleorecords (blue bars
in Fig. 7). These degradation periods, in turn, span the largest eruptions from ca. 1250 to ca.
1850CE, which are associated with the multi-decadal variability of Sahel precipitation over the
past millennium in PMIP4 multi-model experiments (Villamayor et al. sub.).

**4.2.1 Steps in the degradation of the climate and the environment in the Sahel**

Three major steps are identified:
-   The first dramatic environmental degradation occurred between 1290 and 1350 CE
(event 1), i.e., ca. 50 years after the first warning signal and lasted about 60 years. Dust
fluxes to the ocean, which had stabilized during the medieval warm period, increased
(Mulitza et al. 2010) whereas lake levels dropped in the interdunal depressions in the
western Sahel leading to the salinization of the waters (Lézine et al. 2019).
-   The second stage in the degradation of environmental conditions occurred ca 1600CE
(event 3). The environmental degradation was common to the entire Sahel (Bal,



Mboro, St Louis) while corresponding to a major collapse of the forest galleries at
Mboro. Here also, a time lag of ca. 50 years can be observed between the onset of a
drought phase and the response of the vegetation.
-   The ultimate environmental threshold is recorded ca 1800CE (event 6). It resulted in
the widespread lowering of lake levels, the massive contribution of dust to the ocean,
and the irreversible destruction of forest galleries in the western Sahel in response to
an abrupt drop in rainfall ca 1800CE, already observed by Carré et al. (2019) in the
Saloum river delta. By accounting for a catastrophic decrease in precipitation of -0.6
mm day$^{-1}$ over 50 years in our model experiments, this climatic tipping point related
to closely spaced large volcanic eruptions (starting with Laki eruption in 1783 CE
followed by the eruptions cluster over the 1809-1835 CE period including the 1815
Tambora event), at the origin of the modern environmental conditions in the Sahel,
was twice as large as the early warning signals identified at the end of the MCA.
Our data-model comparison suggests that there was a time lag of several decades
(typically 50 years) between the climate signal and the environmental response. If this
time lag is highly probable, its duration and origin require further investigation. It may
indeed result from the resilience of plants to climate change but we cannot exclude the
memory effect of aquifers already observed by Aguiar et al. (2010) that may induce a delay
between the climate signal and its effects on ecosystems. The uncertainty associated with
the ages, whether it comes from the data or from the modelling, can also play a role by
increasing or reducing this response time.
**4.2.2 The Savannah zone:**
As the ITCZ moved to more southerly latitudes, some of the drought events reconstructed in
the Sahel had a major impact in the Savannah zone. Here, data is particularly sparse and, as in
the Sahel, changes in vegetation are hardly distinguishable in these already highly degraded
environments, such as at Lake Sélé (Salzmann et al. 2005). It is at Lake Petpenoun (Catrain
2021) that the evidence is the clearest due to the presence of a gallery forest and pronounced
hydrological changes at the core site.
We find that the last step of degradation of the savannah vegetation occurred during event 3
also observed in the Sahel. Events 2 (1447-1493CE), 4 (1643-1657CE) and 5 (1691-1707CE)
correspond only to phases of hydrological degradation that are not reflected in the regional
vegetation. Data are still too rare to generalize this observation to the entire Savannah zone
and could only account for local conditions.
**5. Conclusion**
Despite the uncertainties associated with data scarcity and heterogeneity, our study shows a
remarkable agreement between the data and our past1000 model experiments for
reconstructing the climate and environmental changes in response to natural forcing that
characterized the LIA in western Africa. It highlights a North-South contrast between the
dryness of the Sahel and the humidity of the equatorial zone. Despite the major difficulty
related to the type of vegetation at play in the Sahel and the Savannah zone already degraded
since the end of the AHP, major steps in the degradation of the environment can be identified.
Our most remarkable results consists in (1) the identification of two early warning signals at



1170 and 1240CE , i.e. prior to the progressive LIA drying of the Sahel that lead to the climatic
tipping point at 1800-1850CE. This tipping point marks the setting of arid conditions (the driest
condition since 850CE) which still persist today; (2) the identification of abrupt drought events
which punctuated the LIA, the most important of them has impacted both the Sahel and the
Savannah zone ca. 1600CE. The consistency between proxy records and our model
experiments suggests a strong role of large volcanic eruptions in shaping Sahel environmental
changes over the pre-industrial millennium. Further work relying in large ensembles of climate
and vegetation models will help assess such hypothesis.
**Code availability**
The IPSL-CM6A-LR model code used in this work was frozen (version 6.1.0) and subsequently
altered only for correcting diagnostics or allowing further options and configurations. Versions
6.1.0 to 6.1.11 are therefore bit-reproducible for a given domain decomposition, compiling
options and supercomputer. LMDZ, XIOS, NEMO and ORCHIDEE are released under the terms
of the CeCILL licence. OASIS-MCT is released under the terms of the Lesser GNU General Public
License (LGPL). IPSL-CM6A-LR code (version 6.1.0) is publicly available through Apache
Subversion (svn) control system, with the following command lines under Linux: svn co
http://forge.ipsl.jussieu.fr/igcmg/svn/modipsl/trunk modipsl; cd modipsl/util; ./model
IPSLCM6.1.11-LR (IPSL-CM model development team, 2021). The mod.def file provides
information regarding the different revisions used, namely (1) NEMOGCM branch
nemov36STABLE revision 9455; (2) XIOS2 branchs/xios-2.5 revision 1873; (3) IOIPSL/src svn
tags/v224; (4) LMDZ6 branches/IPSLCM6.0.15 rev 3643; (5) tags/ORCHIDEE20/ORCHIDEE
revision 6592; (6) OASIS3-MCT 2.0branch (rev 4775 IPSL server). The login and password
combination requested at first use to download the ORCHIDEE component is "anonymous"
and "anonymous". We recommend referring to the project website,
http://forge.ipsl.jussieu.fr/igcmg_doc/wiki/Doc/Config/IPSLCM6 (IGCMG, 2022), for a proper
installation and compilation of the environment (version 6.1.10).
**Data availability**
Pollen data are available on the African Pollen Database website:
https://africanpollendatabase.ipsl.fr. The other paleo-data are from the literature.
The IPSL-CM6A-LR model data and pre-processed model and proxies datasets used in this
study are available at: https://doi.org/10.5281/zenodo.7003853
**Author contribution**
AML and MK designed the study. MK performed the IPSL-CM6A-LR model past1000
simulations and JV the simulations analysis. MC and AML collected and analyzed the data.
AML prepared the manuscript with contributions from all co-authors.
**Competing interests**
The authors declare that they have no conflict of interest



**Acknowledgements**

This work contributes to the ACCEDE ANR Belmont Forum project (18 BELM 0001 05). This work was undertaken in the framework of the French L-IPSL LABEX and the IPSL Climate Graduate School EUR and benefited from the FNS "SYNERGIA EffeCts of lArge voLcanic eruptions on climate and societies: UnDerstanding impacts of past Events and related subsidence cRises to evaluate potential risks in the future" (CALDERA) project under French CNRS grant agreement number CRSII5_183571 – CALDERA. MK acknowledges support from the HPC resources of TGCC under the allocations 2020-A0080107732 and 2021-A0100107732 (project gencmip6) provided by GENCI (Grand Equipement National de Calcul Intensif) and 2020225424 provided by PRACE (Partnership for Advanced Computing in Europe). This study benefited from the ESPRI computing and data centre (https://mesocentre.ipsl.fr) which is supported by CNRS, Sorbonne Université, Ecole Polytechnique and CNES as well as through national and international grants. Thanks are due to the African Pollen Database for data access. AML, MC and JV were funded by CNRS, and MK by IRD. We acknowledge the World Climate Research Programme's Working Group on Coupled Modelling.

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
