# Peer review of "Using data and model to infer climate and environmental changes during the Little Ice Age in tropical West Africa"

_Climate of the Past, 2022_

## Community Comment (CC1)

Reviewer 1

The manuscript presented by Lézine et al deal with the climate changes experienced in tropical West Africa during the last millennia. The manuscript first makes use of a number of paleo-records in the study area to define two new indices able to quantify the hydrological and vegetation context. The goodness of these paleo records is first evaluated by computing two multi-proxy indices which are validated against instrumental data for the period 1840-present with good results. Then, the authors make use of modelled data (from 850 to 1850) to characterise the precipitation in pre-instrumental period, subsequently discussing the relation between the modelled climate and the observed variability of the paleo-records, offering an interesting discussion and relevant results.

The 1erforme well written and in general it is clear. Personally, I find this work quite interesting, as it deals with a region still poorly characterised because of the sparsity of instrumental or even proxy records. Therefore, I recommend its publication in Climate of the Past.

I however have some concerns (general comments) that, if addressed, will probably improve the clarity of the manuscript :

The most important is related with the methodology used to homogenize the paleo data in section 2.1. As far as I understand, the method 1erform based on rescaling each individual paleo record (table 1) to a common 6-level scale. However, the details of this conversion are not explained in the manuscript making it impossible to know how this index is ultimately computed (this is essential at the time of evaluating the goodness of the original data or even to allow reproducibility). In my opinion, the authors should describe a little more the way this rescaling has been performed.

This paper is based on a *qualitative* description of regional environmental and climate conditions. As is now shown in the supplementary figure (see below), the index synthesizes data from different proxies types (e.g. pollen percentages or influxes, diatom percentages…) from which the main features indicative of aridity are extracted based on a step-scale.
The supplementary figure shows that we relied on proxy such as salt-tolerant diatoms concentration (at Mboro site) which allows identifying the development of aridity based on the salinity levels of lake waters. We also relied on several pollen taxa (such as at Petpenoun), where the development of aridity is deduced with the transition from plants typical of open water (Nymphaea) to plants typical of lake edge (ferns).
The purpose of these step-scale indexes is to homogenise the information provided by the heterogeneous and complex original data sets. The step-scale is built to capture the major transitions to allow distinguish the signal from the noise.
The figure illustrates the method with an example from each of the major vegetation zones considered in the paper: the sahel (Mboro Baobab), the savanna zone (Petpenoun), the mountain forest (Bambili) and the lowland evergreen forest (Ossa). The index is drawn manually from original data.

[Figure]

Another question is related to the reason why the authors have limited their study period to 850-1850. I'm not familiarised with the past1000 experiment data but ending in 1850 most probably indicates that the past1000 experiment was conceived to model the pre-industrial era. However, if possible, 2erform be extremely interesting that the modelled precipitation series were extended to present time. This would allow to compare the model results with the instrumental ASWI (figure 2 of the manuscript) and, providing the result is good, it would add a lot of confidence to the results. Anyway, I would like to stress that I find figure 2 very interesting as, beyond some indirect evidence, the humid period described by the ASWI between 1840-1890 had not be confirmed by independent data up to now.

We restricted our model-data comparison to the pre-industrial past1000 period as the transition between the past1000 and the historical period since 1850 marks a large change in the signal-to-noise ratio between the magnitude of the external forcing and the internal variability. Any model-data comparison allowing the validation ASWI index and the model skills over the historical period would require a different strategy. A previous study by Villamayor et al, 2018 relying on an ensemble of experiments with the atmospheric component of the IPSL coupled model with imposed observed sea surface temperatures has indeed shown a good consistency with the ASWI index regarding the late nineteenth-Century

Sahel humid period suggesting that sea surface temperatures in the Atlantic basin played the dominant role (https://doi.org/10.1175/JCLI-D-18-0148.1 ).

It is also worth mentioning that coupled ocean-atmosphere models display however large uncertainties over the historical period related to the emergence of anthropogenic forcing (GHG, tropospheric aerosols, land-use changes). Previous work attributes this uneven CMIP6 model response to anthropogenic forcing to dynamical changes (Phal et al. 2017) linked to the Northern Hemisphere and tropical Sea Surface Temperatures, identified as important sources of uncertainty for the simulated Sahel rainfall over the historical period (Park et al. 2015; Zhang and Li 2022).

Apart of these questions, there are some minor aspects that could help to clarify the text (specific comments) :

Lines 31-32. The west African monsoon is not only driven by land-sea contrast, but it is also a consequence of the migration of the ITCZ (see for example Gagdil et al. https://doi.org/10.1007/s12040-017-0916-x).

We agree with the reviewer as we show in our study that the ITCZ migration is a key mechanism driving the West African monsoon variability over the last millennium. We corrected the sentence and added the suggested reference on lines 31-33 of the revised manuscript.

Line 50. I consider that this manuscript is not a "review" but a "research".

corrected

Table 1 : Maybe expressing the latitude and longitude in sexagesimal form will be clearer.

The latitude and longitude are expressed in decimal form in most of the international databases and geographic information systems.

Line 97 : In my view, the validation performed is not indicating that the methodology is "realistic" but instead, it is testing the similarities between the paleo-data and the instrumental ASWI.

corrected

Figure 1. The blue arrows are a little difficult to see where the underlying colour is also blue.

Redrawn in dark grey

Line 180. I believe that the way the past1000 index is constructed should be more explained.

Following this suggestion, we have made an effort to simplify and clarify the methodology employed to calculate the past1000 index of Sahel precipitation. Please note the changes in the revised manuscript in lines 239-250:

"Then, to characterize the simulated Sahel rainfall multidecadal variability over the past millennium and contrast to the reconstructed environmental series, an index is performed as the 10-year low-pass-filtered Sahel precipitation anomalies in the rainy season from past1000 simulations. Seasonal precipitation anomalies from July to September (JAS), relatives to the piControl climatology, are area-weighted and averaged across the Sahel region (red box in Fig. 3A), then filtered with a 10-year moving mean. An ensemble-mean index is also performed to highlight the forced component of the Sahel multidecadal variability in response to natural forcings that are common to the three past1000 members, such as the effect of large volcanic eruptions, in contrast to the internal variability."

Figure 4. I'm sure that presenting such amount of series in a single figure is not easy, but it is quite difficult to interpret some of the y-axis scales in this figure. For example (not the only case) in figure 4A "Jikaryia" the axis is scaled by not consecutive values (3, 1.5, 2 , 0.5). Please clarify.

corrected

Line 326. Please indicate the methodology used to compute statistical significance.

Please note the correction in the caption of Figure 6 in the revised version of the manuscript: "Stippling indicates full agreement across the three past1000 members on the sign of the represented difference."

Line 352. The local term "Heug" could be unknown by readers not familiarised with the climate of this region. Please explain.

done

---

## Author Comment (AC1)

In this manuscript, Lezine and colleagues present a synthesis of palaeorecords representative of the WAM covering the period between 850 – 1850 CE. They compare these data with recent simulations over the same period. The paper is interesting, well-written, and summarises the region's state-of-the-art quite nicely. As such, I recommend its publication in Climate of the Past, providing some clarifications of some technical elements supporting the study (see below). Most data seem to be represented with some indices. How these are constructed is rarely described, making reading and interpreting the figures quite difficult. The authors mention a 6 levels scale, but many plots display decimal values (e.g. 2.5 for GeoB9501 in fig. 4). Overall, the manuscript would greatly benefit if all the technical details of this study were better described.

We thank the reviewer 2 for his positive appreciation of our article. His remarks help to clarify and improve the manuscript. We answer them point by point.

(1) The major question concerns the methodological aspect (the elaboration of indices of paleoenvironmental change). We have answered in detail a similar question from reviewer 1. We reproduce our answers here.

**"This paper is based on a *qualitative* description of regional environmental and climate conditions. As is now shown in the supplementary figure (see below), the index synthesizes data from different proxies types (e.g. pollen percentages or influxes, diatom percentages…) from which the main features indicative of aridity are extracted based on a step-scale.**
**The supplementary figure shows that we relied on proxy such as salt-tolerant diatoms concentration (at Mboro site) which allows identifying the development of aridity based on the salinity levels of lake waters. We also relied on several pollen taxa (such as at Petpenoun), where the development of aridity is deduced with the transition from plants typical of open water (Nymphaea) to plants typical of lake edge (ferns).**
**The purpose of these step-scale indexes is to homogenise the information provided by the heterogeneous and complex original data sets. The step-scale is built to capture the major transitions to allow distinguish the signal from the noise.**
**The figure illustrates the method with an example from each of the major vegetation zones considered in the paper: the sahel (Mboro Baobab), the savanna zone (Petpenoun), the mountain forest (Bambili) and the lowland evergreen forest (Ossa). The index is drawn manually from original data."**

[Figure]

(2) Decimal values were used to identify minor changes in the paleoenvironment. (added in the text)

Specific comments

L76: Is this the mean resolution of 100 years? Or the highest time between two consecutive samples should be 100 years?

It is the maximum time interval between two consecutive samples

L78-81: I understand this part perfectly, but I do not like the use of the term 'degraded', which is biased towards human perception. Plants or animal species that live in 'degraded' (as defined here) environments would probably not call it that way. A more objective description, from most arid to humid conditions, seems more appropriate.

The sentence has been modified accordingly:

"in order to evaluate the relative amplitude of the environmental/climate change, we build a 6-point scale ranging from 0, corresponding to the most arid environment (e.g., drying of lakes, salinization of water, increase of dust transport, opening of the vegetation cover) or the driest climate, up to 6, which

refers to the most humid environment (e.g., high lake level, fresh water, dense vegetation cover) or the wettest climate. Decimal values were added to identify minor changes in the paleoenvironment ».

L97-113: I think a description of how the ASWI is calculated is warranted. Not in detail, because it has been published elsewhere, but with sufficient information to avoid checking the Gallego et al. 2015 ref.

The sentence has been modified accordingly:

« The ASWI is based on JAS wind direction data (i.e. the persistence of the low-level south-westerly winds) from historical measurements available since 1839 in a region over the Atlantic, close to West Africa (29°W–17°W, 7°N–13°N) . The ASWI is strongly correlated with the observed Sahel precipitation since 1900 and is, therefore, presented as a good indicator of its variability. »

L140: Why a reference to ENSO here? It doesn't seem to be contributing to the rest of the study.

We think it is always good to discuss the model skills regarding ENSO as this mode has a strong contribution to Sahel rainfall variability. Having a reasonable representation of this feature would hence give more confidence in the robustness of our results.

L141: Define the acronym AMV

We rephrased as bellow :

The spatial pattern of the  Atlantic Multidecadal Variability (AMV; Deser et al, 2010) teleconnection in the Pacific is consistent with observations but the tropical Atlantic variability is relatively weaker.

Fig. 3: This may be a problem with the preprint, but the axes and colour scale labels are difficult to read. This comment also applies to most figures.

The fonts have been made bigger

L173-175: Replace 'slightly' with a measure in distance or degrees. The northward expansion seems to be several degrees south of where it should theoretically go. Then discuss why this is acceptable.

We rephrased as bellow :

"However, the northward shift of maximum rainfall over the Sahel during the rainy season is underestimated by the model by about 4° (the model's maximum in August is at ~7ºN and the observed one at 11ºN). As a result, the climatological rain belt over West Africa is slightly more constrained to tropical regions compared to observations and with a dryer Sahel on average. "

L180: An index is calculated or derived, not performed. What does this index measures? What do you do to the first and last 9 samples when computing the moving average since they do not meet the criterion of 10 samples for the moving average? Are you reducing the length of the record? (No wrong answers here, but the methodology needs more clarity).

Following the reviewer's recommendation, we replace "performed" with "calculated". We also specify that the endpoints of the low-pass-filtered index are truncated to keep the full 850-1849 time range. This is taking only the elements that fill the centred 10-year moving window at the endpoints to calculate the mean. Please note the following changes in the manuscript:

"Seasonal precipitation anomalies from July to September (JAS), relative to the piControl climatology, are area-weighted and averaged across the Sahel region (red box in Fig. 3A), then filtered with a 10-year centred moving mean with truncated endpoints (i.e., only averaging existing elements within the 10-year window)."

L179-189: A better description of how the index is calculated and what it means is warranted here.

The text has been modified accordingly:

"Then, to characterize the simulated Sahel rainfall multidecadal variability over the past millennium and contrast to the reconstructed environmental series, an index is performed as the 10-year low-pass-filtered Sahel precipitation anomalies in the rainy season from past1000 simulations. Seasonal precipitation anomalies from July to September (JAS), relatives to the piControl climatology, are area-weighted and averaged across the Sahel region (red box in Fig. 3A), then filtered with a 10-year centred moving mean with truncated endpoints (i.e., only averaging existing elements within the 10-year window). An ensemble-mean index is also performed to highlight the forced component of the Sahel multidecadal variability in response to natural forcings that are common to the three past1000 members, such as the effect of large volcanic eruptions, in contrast to the internal variability."

Fig. 4: What do the blue shades indicate? And how were they determined? Also, what are the numbers of the y-axes? ASWI values?

The figure caption has been improved and this sentence has been added:

« The shaded bands indicate the transition period between the medieval climate anomaly and the Little Ice Age (1250-1450CE light shading) and the LIA (1450-1850 dark shading) »

Results hydrological records: I am struggling to find commonalities within the groups of hydrological records, except for the Sahel region, where a general trend toward drier conditions seems to be consistently reconstructed. Perhaps the authors could run some statistical analyses to extract the trends shared by the records and limit the impact of the 'local' signals.

We totally agree with this observation. Vincens et al (1999) already noted the importance of the local hydrogeological context in the response of lakes and wetlands to climate change, which explains the complexity of the records in the equatorial lowlands. Nevertheless, figure 6 shows that, despite a great hydrological variability, the equatorial region was overall "wet" during the LIA, in contrast to the previous period.

Results pollen records: How are pollen records summarised to one single curve? They seem to be plotted against – broadly speaking – the same scale as the hydrological records. Did the authors convert every pollen sample, a high-dimension type of data, to one single value by hand? More details are definitely needed here.

See above response to reviewer 1

In addition, hydrological and pollen records from the same site are, more often than not, quite different.

The lowland equatorial forest is not significantly sensitive to discrete variations in hydrology in a context of prevalent humidity compared to the wooded grasslands and woodlands of the Savannah zone and the Sahel.

The text has been revised as follows:

« The equatorial lowlands are characterized by contrasting hydrological situations (Fig. 4E) reflecting the diversity of local hydrogeological settings » (3.1)

"The forest cover remained roughly unchanged in the central forest massif (Mopo Baï, Bamba Yanga, Goulalougo, Fig. 4I). In the western regions by contrast (Ngamakala, Kitina, Lac Ossa, Nguène and Kamalété), the forest gradually developed since 1250-1350CE in spite of the discrete hydrological fluctuations. (3.2)"

Fig 6: The signal obtained from the data seems consistent, despite my comments above. Maybe this suggests that their representation in Fig. 4 could be somehow improved.

Another figure similar to Fig. 6 but for the MCA would be pretty important here to see if the changes captured by the data represent the effect of MCA to LIA transition or if it is something else.

This is not the purpose of our article. A synthesis on MCA throughout Africa has already been published by LÜNING, S. et al. Hydroclimate in Africa during the medieval climate anomaly. *Palaeogeography, Palaeoclimatology, Palaeoecology*, 2018, vol. 495, p. 309-322.

Fig. 7A: How are these 'regional' curves derived?  [I found the explanation later on lines 406-407. It would still be good to add it to the caption]

The regional curves are the mean of the curves for each site in the two regions considered

---

## Author Response (AR2)

Paris, 16/12/2022

Dear Editor,

We thank you for your last review (minor revision) of our article:xxx. We have done our best to  address the reviewers questions and requests and updated the manuscript accordingly. We have deposited the processed datasets at the following DOI https://doi.org/10.5281/zenodo.7003853. The raw data are available either in databases or in the literature. A Supplementary file is now included  which illustrates our methodology for the 2 categories of data used. We believe that our approach, based on our expertise of more than 40 years regarding hydroclimate and paleoecology in Africa, is honest and realistic. It offers the possibility for readers unfamiliar with paleodata to have a direct and simplified reading of the environmental change of the last millennium.  We also want to mention that such an approach was cross-validated by instrumental  data and independant published datasets such as the semi-quantitative index developed by Nicholson, which also relies on  indirect data (i.e. narratives or archaeological findings) that are discontinuous and widely dispersed geographically and chronologically. We hope that the manuscript is now acceptable for publication.

Yours, Sincerely

Anne-Marie Lézine

**Reviewer 1**

The abstract presents the second result using unnecessarily strong words: irreversible? Nothing says these changes cannot be undone. Dramatic? At most, the records drop by 2 units of the qualitative scale, 2 out of 6. And they are never reaching 0. These words do not add anything to the study, and it reads as if the authors were trying to oversell their results. This is not necessary and detracting.

The abstract has been modified accordingly.

However, the reviewer can observe that the zero value (bare soil) is reached at Yoa. This value cannot be reached for the hydrological record since the lake or marine series never show complete drying of lakes, wetlands and rivers. Conversely, the maximum value (6) is reached in the tropical forest. The intermediate values are 1: steppe/grassland; 2 wooded grassland; 3: woodland/degraded forest; 4: secondary forest; 5: montane forest (panel F to I). For hydrology and climate, the index shows the evolution from dry (1) to wet (4) with intermediate values showing the gradation between these two extremes (panel A to E)
The legend of the figure 4 has been changed accordingly.

Being not an expert on the regional climate, I would appreciate seeing the convincing explanation about the role of ENSO provided in the responses in the manuscript. Without that background knowledge, the sentence is still floating and not connected to the study. Or is the sentence about the AMV's spatial pattern in the Pacific supposed to make the link? I trust the

authors that everything here is relevant, but I am unsure how. Please make this clearer for uneducated readers.

As discussed in the introduction, it is well known and admited that the West African Monsoon and Sahel rainfall variability during the 20th century was mainly driven by contrasting patterns of sea-surface temperature (SST) anomalies related to both the Atlantic Multidecal Variability and the El Niño Southern Oscillation global teleconnections (Folland et al. 1986; Mohino et al. 2011; Rodríguez-Fonseca et al. 2015), amplified by land surface processes (Giannini et al. 2003; Kucharski et al. 2013). We refer the reviewer to the cited articles in the introduction for more details. It is, in that sense, relevant to discuss the general skills of our climate model in representing these main features to assess the validity of the protocol and confidence we can expect in the robustness of our results.

I am still trying to be convinced by the 6-point scale devised by the authors, but this 'handmade' transformation is bugging me. I would recommend doing this using a mathematical transformation to ensure the correct assignment. Otherwise, I do not see how a proper classification consistency could be reached. Alternatively, I would appreciate seeing an appendix with the summary curve on top of the actual data, similar to the figure provided in the responses, to enable readers to also get a feel for the data supporting these qualitative indices.

The paleodata available are of various types. There are (1) original data (e.g. raw pollen counts) that are available either in the literature or in databases and (2) published data but whose authors do not provide the original counts and whose curves have been re-drawn from published figures. There is no reason to question these published data which have gone through the review process (3) and finally, there are complex published data such as vegetation succession, so that a single curve is not sufficient to encompass the entire local evolution of hydrology and vegetation. This is why it is not possible to use a mathematical transformation as suggested by the reviewer, while the processing and analyses of these 3 groups of datasets collectively by experts can provide a valuable picture of environmental changes in the region thoughout the last millenium. All the indices are avalable at the following link : https://doi.org/10.5281/zenodo.7003853. As requested by the reviewers we also provide the supplementary figure illustrating the method used to deduce the indices from the 3 types of data.

We also want to stress that our methodology follows the procedure developed by Nicholson (Nicholson, S. 1978 Climatic variations in the Sahel and other African regions during the past five centuries. Journal of Arid Environments 1, 1, 3-24). Such semi-quantitative historical data was assessed and cross-validated by Villamayor et al, 2018, relying on the independant reconstruction from Gallego et al, 2015 and with instrumental observations (see figure below from Villamayor, J., Mohino, E., Khodri, M., Mignot, J., & Janicot, S. (2018). Atlantic control of the late nineteenth-century Sahel humid period. *Journal of Climate*, *31*(20), 8225-8240).

**Figure from Villamayor et al, *Journal of Climate*, 2018**

[Figure]

FIG. 4. The bars show the semiquantitative index of Sahel precipitation of the reconstruction of NI12. The lines represent the seasonal ASWI in JAS of GA15 (black) and the observed index of JAS seasonal precipitation in the Sahel (averaged in 17.5°W–10°E, 10°–17.5°N) in observations (green) and in the ensemble-mean simulation (blue). The last three indices have been low-pass filtered with an 8-yr cutoff period and standardized with respect to the observed period (1901–2000). The blue shading is the standard deviation among the 19 members simulated.

We have done our best to answer the reviewer in our previous response. We cannot do better at this stage given the heterogeneity of the data used.

**Reviewer 2**

Please, when possible, include the references leading to the data used in the example presented in the supplement

Done